# How Much Does Water Management Cost? The Case of the Water Market in the Ñuble River of South-Central Chile

Bratian Buzolic [1,2,*], José Luis Arumí [1] and Jorge Jimenez [2]

1    Water Resources Department, CRHIAM Center, ANID/FONDAP-15130015, Campus Chillan, College of Agricultural Engineering, University of Concepción, Av. Vicente Méndez 595, 3812120 Chillán, Chile; jarumi@udec.cl

2    Department of Industrial Engineering, University of Concepción, Edmundo Larenas 215, 4070409 Concepción, Chile; jorgejimenez@udec.cl

*    Correspondence: bratianbusolich@udec.cl; Tel.: +56-9-6196-5487

**Abstract:** Economic and population growth has increased the demand for freshwater worldwide, generating pressure on the environment and creating conflicts among users. Water markets have emerged as a solution for managing this resource, and Chile has been a pioneer in implementing this approach. However, most Chilean water markets are inefficient due to incomplete information, the poor flexibility of the water distribution system, and high transaction costs. This study analyzes the Ñuble River water market and estimates the economic and social costs of its inefficiencies through a methodology based on the marginal profitability of water, which simulates the operation of a perfect market for the Ñuble River irrigation system. Net benefit losses from market inefficiencies were estimated at 7.6 million dollars annually, which is equivalent to a 25% increase in the net returns of the current river water distribution strategy. Losses of economic benefits are even greater as the availability of water flow decreases. This is important considering that in the last decade the water flows of the Ñuble River have decreased by more than 30% compared to their historical average.

**Keywords:** transaction costs; imperfect market; water administration; water market; proportional water distribution

## 1. Introduction

In the last century, population and economic growth have led to an increase in the demand for water, both for human consumption and the production of goods and services [1]. In addition, climate change and land-use changes have led to greater variability in the surface and groundwater supply, resulting in a downward trend in the amount of freshwater available for human consumption and production [2]. This has led to a change in the global paradigm of water management, moving from a focus on supply management [3,4], which seeks to increase freshwater availability through artificial reservoirs, deep wells, and aqueducts, to a focus on demand management [5,6]. The latter approach seeks to increase the benefits of water use through distribution strategies and use efficiency, as happens in countries like Great Britain, Australia, United States, Canada, South Africa, and Chile [7].

In Chile, the institutional policy framework established in 1973 was aimed at creating solid, secure, and tradable water use property rights and facilitating the efficient operation of the market as a water allocation mechanism. Water use rights are private property guaranteed by the Political Constitution of the Republic (PCR). As water use rights are defined as private property, they may be freely transferred, purchased, or leased like any other private good [8,9]. However, this system does not ensure the existence of tangible and intangible infrastructure for the correct operation of the model [10].

The Ñuble River is located in south-central Chile, an area that is undergoing the longest drought on record [11], with more than ten years having passed without average annual rainfall or runoff levels being met. Nonetheless, it has an abundance paradigm

irrigation system that uses water distribution boxes for its canals. This distribution method is a fixed structure based on a critical flow that provides water distribution proportional to water rights. This method is similar to those present in gravitational distribution systems developed in southern Europe, northern Africa, Pakistan, India, and central Asia [12,13]. These distribution systems are rigid and do not allow for transitory changes in water allocation. Therefore, no real benefits can be achieved from a water administration policy based on demand management [14].

This research focuses on the evaluation of the theoretical economic benefits of flexible water allocation in the Ñuble River by simulating a perfect temporary water market, in contrast to fixed water-rights-based allocation. The market was used in the simulation because the country has defined the market as a form of rights management; thus, it would be illogical to use a different scenario such as centralized allocation. In addition, it was used in theoretical perfect market conditions, since, as the market is an alternative subject to the will of the participating parties, it must be approached in its theoretical perfect conditions to allow impartiality. This study may be of global interest since it addresses water allocation problems [15] using a supply management approach, which includes a water reservoir that is under technical assessment. However, the lack of flexibility of its irrigation infrastructure could significantly reduce the potential benefits of this project. This analysis may be useful for comparison with other irrigation systems with similar water allocation infrastructure in Chile and other countries.

The purpose of this study is to contribute to the literature by providing a methodology for estimating the opportunity cost of maintaining a fixed traditional irrigation canal system under climate variability. This methodology was applied as a case study of the Ñuble River water market in order to estimate the economic and social costs of the current water distribution system. Understanding the implications of the current water management approach will allow decision-makers to justify the costs of upgrading the current irrigation infrastructure for more efficient use of water in areas with limited access to this resource.

### 1.1. Chilean Water Management

Water management in Chile is characterized by self-government through water user associations (WUAs) [9,16,17] that engage in canal maintenance, operation, conflict resolution, and political–legal representation through collective choice. The first law to regulate them was passed in 1908 [18], and the task of water reallocation has predominantly been carried out by water markets, whose regulating framework was updated in the 1980s. Currently, water administration is governed by the 1980 Political Constitution of the Republic of Chile (PCR) and the Water Code of 1981 (WC), both adopted during a military government. The Water Code establishes water as a national good of public use, for which the government can grant concessions called water use rights (WUR), which allow this public good to be used for private purposes. These rights are granted in perpetuity and independent of how they will be used and the land or location in which they were initially granted [8]. Meanwhile, the PCR, in Article 19 Number 24, establishes that "Private water rights, recognized or constituted pursuant to law, shall grant their holders ownership over them." This principle establishes private ownership over water rights, which is strongly protected and guaranteed by the constitution, providing freedom to private parties with limited regulation and a strong and costly [19] judicial system for conflict resolution [4,20].

It worth mentioning that neither the WC nor the PCR mandate or establish a water rights market. Rather, they aim to establish laws and requirements for water markets to emerge spontaneously as a result of a private initiative [5]. In addition, water markets lack any organization that would ensure the minimum conditions necessary for development into competitive markets (perfect information, low transaction costs and market planning, among others). This situation has generated asymmetric information, large differences in negotiating power and price dispersion, which are indicators of an inefficient market for allocating a valuable resource [10].

A regulating framework of the Chilean water market uses a free-market approach with foundations in Coasean economic principles [21]. The Coase theorem [22] applied to water assumes that independent of the initial allocation of property rights, the optimal allocation of water rights will be achieved through the market. The externalities will be minimized because parties will naturally negotiate the most mutually beneficial result [23]. This occurs assuming a market without transaction costs, implying perfect information and symmetrical negotiating powers. However, in the absence of these elements, transaction costs are relevant and would be required to obtain them. Considering that water is a common (shared) resource for water users and non-users, conflicts, and environmental degradation may occur when individual users act on the resource according to their own self-interest [24]. This condition also leads to sub-optimal allocations of a common resource such as water. Therefore, unless the negotiation is carried out with the entire collective, an economic optimum will not be achieved.

*1.2. Transaction Costs*

In a water market, there may always be transaction costs [25], which can be physical or intangible costs. The former result from changes in the distribution infrastructure, and the latter are manifested in monitoring and control on the one hand and information-seeking and negotiating on the other [26]. The cost of changing water allocation in a rigid irrigation network is high due to the lack of adjustable flow dividers [27], which must be rearranged to allow water transactions among users. With the new technologies of telemetry and remote control of automated gates, the costs of changing water allocation can be significantly reduced.

Monitoring and control costs are comparatively low due to the institutional arrangement of self-government among WUAs [28]. However, information-seeking costs are high because information is limited and difficult to access [19]. Currently, the Public Water Registry (CPA, for its initials in Spanish) of WUR owners remains incomplete [23,29], and the water transaction pricing through the Real Estate Registry Office (CBR, for its initials in Spanish) is confusing and inaccurate in terms of posting the real fare of water share transactions [4,30]. In addition, there is no organization that brings together water buyers and sellers, making it necessary to obtain this information through informal means [31,32]. All these factors increase negotiation costs due to the need to strengthen legal backing in high-uncertainty scenarios.

*1.3. Study Area*

The Ñuble River irrigation system (ÑRIS) is located in the Ñuble Region, approximately 400 km south of Santiago, Chile (see Figure 1). The main economic activity there is agriculture, and the region has a long-established water user association (WUA). A significant irrigation project is currently under development in the region, which includes the construction of the 600-hm$^3$ Punilla reservoir. This irrigation project will help expand the productive capabilities of the area, since crops with the greatest market value, including fruit trees, are water-intensive. Unfortunately, neither the irrigation water management system nor the distribution infrastructure in the area is prepared to distribute the necessary water to meet all demands. Therefore, farmers will not be able to efficiently manage their water or exchange it in the current water market. Thus, the irrigated lands of the region will not reach their full productivity potential.

The problems with the current water management system in the ÑRIS include: (i) missing information on streamflow delivered to end users; (ii) incomplete information on the water market, as there is no institution that keeps track of water prices, sellers, and buyers, and (iii) rigid distribution infrastructure that allocates water proportional to the streamflow of the river (flow dividers). This distribution infrastructure generates high transaction costs and does not allow significant changes in distribution and water market operation. Meanwhile, the lack of information on water transactions discourages farmers from actively participating in the water market [33,34] and creates conflicts among water users.

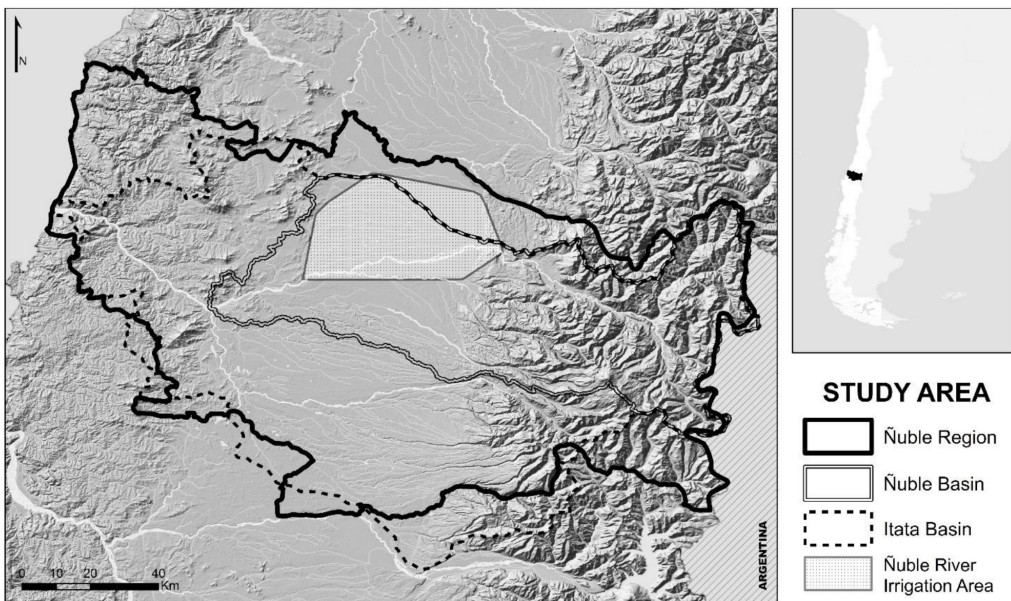

**Figure 1.** Study area. Ñuble River irrigation system (ÑRIS) (white area), Ñuble River basin (double black solid line), Itata River basin (dotted line) and Ñuble Region (black solid line).

The Ñuble River has a pluvio-nival regime with a streamflow that exceeds $100 \text{ m}^3 \text{ s}^{-1}$ in winter and spring. In summer and autumn, when more water is in demand for irrigation, the flows are reduced to less than $30 \text{ m}^3 \text{ s}^{-1}$. The climate of the area is influenced by oscillatory weather patterns, and the El Niño and La Niña phenomena (ENSO) generate interannual variability in temperatures and rainfall. The Pacific Decadal Oscillation (PDO-IPO) causes an inter-decadal variability in precipitation and temperature [35]. These changing patterns also require changes in seasonal optimal water allocations, with the consequent need for reallocation.

The Ñuble River canal network is a group of 51 canals that draws water from the Ñuble River, providing access to water to 5320 end-users in the irrigation zone. The canal network has a total length of 13,493 km and irrigates an area of about 60,000 hectares distributed among the Coihueco, Chillán, Ñiquén, San Carlos and San Nicolás communities. The Ñuble River Water Board currently manages the water supply of the canal network, through which the streamflow of the river is distributed via proration proportional to the water shares owned by users in each canal; there are 21,221 permanent irrigation water shares. One water share corresponds to a fraction of the streamflow of the river, meaning that the equivalent streamflow is variable, fluctuating between its legal maximum of $5.14 \text{ L s}^{-1}$ and close to $1.0 \text{ L s}^{-1}$ in late summer. However, the streamflow defined by the National Irrigation Commission (CNR, for its initials in Spanish) corresponds to a water share with an irrigation safety net of 85%, equal to $1.82 \text{ L s}^{-1}$. A share is not the same as a WUR, although one depends on the other.

At their nodes (Figure 2), the canals have proportional flow dividers, which are critical flow hydraulic structures that divide the incoming variable streamflow of the canals into different outlets at a fixed proportion, according to water rights (Figure 2b). The main advantage of flow dividers is that they allow water to be distributed in a way that is directly proportional to the streamflow of the canal and the rights of each user. However, they do not allow temporary distribution adjustments. This rigid distribution infrastructure becomes obsolete in periods of scarcity because the users cannot manage or exchange water when they do not need it. This is a significant obstacle to implementing a water market and improving the efficiency of water use among end-users.

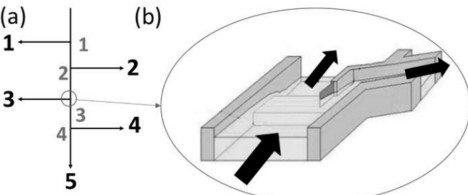

**Figure 2.** Proportional flow dividers. (**a**) Flow divider location in a canal; the blue numbers represent the nodes and the black numbers the users. (**b**) Flow divider diagram.

## 2. Materials and Methods

The exchange of water between lower- and higher-value uses may maximize benefits and the market is the mechanism through which exchanges are made possible. The actions needed to execute these exchanges entail transaction costs, which decrease the benefits of exchange and even prevent exchanges in some cases. The theorical framework is shown in Figure 3 as the aggregate supply and demand model [36].

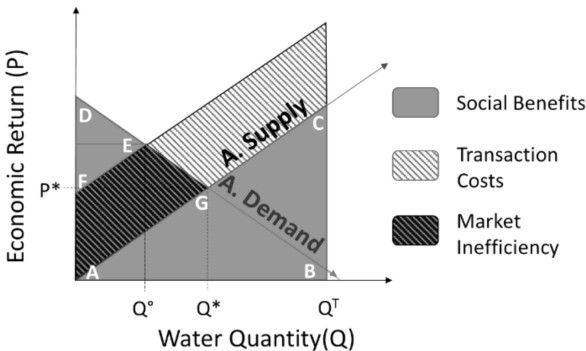

**Figure 3.** Water economic model. Relationship among water volume (Q), economic return (P), transaction costs, social benefits, and market inefficiency. The superscripts *,° indicate points of equilibrium, and the superscript $^T$ represents the maximum water availability for the season. The letters A-G represent vertices of zones or sub-zones of the market that will be analyzed later. Adapted from [36].

### 2.1. Estimation of Benefits

The benefits were estimated based on the marginal profitability of water use in irrigated crops, in this case, those in the Ñuble River irrigation system. Two scenarios were simulated: a perfect temporary water market (WUR lease) for the ÑRIS (ABCGD area; see Figure 3) and water distribution based on non-market rights (ABC area; Figure 3). Marginal crop profitability was calculated as:

$$\pi_t^i \left( X_t^{i*} \left( W_t^i, L^i, R^i, P^i \right) \right) = P^i \times X_t^{i*} \left( W_t^i, L^i, R^i, P^i \right) - R^i \times X_t^{i*}. \tag{1}$$

where $i$ collects the crops present in the ÑRIS, the counter $t$ places the simulation in different water availability conditions according to the fluviometric records of recent years (1986–2016), $\pi_t^i$ represents the profitability of crop $i$ in year $t$, $X_t^{i*}$ is the optimal production proportions vector (benefit maximization), $W_t^i$ is the quantity of water available for crop $i$ in year $t$ ($m^3$) and $L^i$ is the quantity of productive land available for crop $i$ fixed by the VII National Agriculture and Livestock Census of Chile, while $R^i$ is the input value vector ($\$$) and $P^i$ is the price of the crop ($\$$), both values extracted from 2013 Office of Agrarian Studies and Policies (ODEPA) Production Cost reports.

The total water availability for each year ($TW_t$) is equal to the cumulative volume in the river for the irrigation season, as shown in Equation (2).

$$TW_t = \sum_i W_t^i = min\left(\sum_j Qm_j^t \times 86,400 \middle| \sum_i CW_t^i \times L_i\right) \tag{2}$$

where $j$ represents the days of the season between 1 October and 30 April and $Qm_j$ indicates daily mean streamflows on day $j$ at the San Fabián gauging station according to the General Water Directorate (DGA, for its initials in Spanish) in year $t$. $CW_t^i$ is the water demand per crop, which was calculated as the ratio of net water demand and application efficiency [37]. Net water demand was calculated as the difference between the actual evapotranspiration [38] and effective precipitation. The application efficiency was obtained from an agrological study of the zone [37]. Neither groundwater inputs nor the effects of the conveyance efficiency of the canals were included in the model since conveyance losses recharge the river system [39], counteracting their influence on net water availability. Thus, water availability for each crop $i$ in the rights-based distribution scenario will decrease in proportion to its demand in relation to the relative water scarcity in year $t$, as shown in Equation (3).

$$W_t^i = \left(\frac{TW_t}{\sum_i CW_t^i \times L_i}\right) \times \left(CW_t^i \times L_i\right) \tag{3}$$

On the other hand, the availability of water for each crop in the perfect seasonal water market scenario is equal to the maximized benefits of the model shown in Equation (4). There are no constraints on canal capacity, soil or farm characteristics since the current pattern of land use remained fixed. Thus, the maximum amount of water transported in each canal did not increase, and the conditions required for each crop remained the same.

$$Max \; Z_t = \sum_i \pi_t^i \tag{4}$$

Subject to:

$$\sum_i W_t^i \leq min\left(\sum_j Qm_j^t \times 86,400 \middle| \sum_i CW_t^i \times L_i\right), \forall \; t \tag{5}$$

$$W_t^i \leq CW_t^i \times L_i, \forall \; i, \; t \tag{6}$$

$$W_t^i \geq 0, \forall \; i, \; t \tag{7}$$

The model assumptions are as follows: (i) there is an ability to physically transport water and organize its distribution, (ii) WUR distribution is proportional to the total water demand of each crop, including the irrigation surface of the field, (iii) crop prices do not vary with production differences in the ÑRIS, (iv) farmers have access to a perfect market and hydrological information, (v) all WUR (21,221 shares) are properly registered, (vi) the profitability of water use behaves linearly with its quantity and (vii) water quality and hydrological conditions are not modified.

### 2.2. Transaction Costs

Transaction costs have a significant impact on water market outcomes [25,40,41] as they prevent possible exchanges since the benefits are lower than the transaction costs, as is the case between Q° and Q* in Figure 3. The exchange benefits are equal to the difference between the benefits of water use minus the transaction costs (Q < Q°). Overall losses due to transaction costs are equal to the area of the trapezoid AGEF, whose width depends on the magnitude (value) of the transaction costs. This is common in infrastructure adaptation and public policies, such as taxes or offsets [15,42]. Thus, the benefits of the

water market were estimated by including transaction costs in the model as variable and fixed components, respectively.

Water traded in a perfect water market was calculated as the difference between the water that each crop would have received in a rights-based distribution (Equation (3)) and the water potentially received in a perfect market scenario (Equation (4)), as shown in Equation (8). $MP_t$ is the market return, $WM_t^i$ is the amount of water corresponding to each crop resulting from market-based distribution (Equation (5)), $WR_t^i$ is the amount of water corresponding to each crop under a rights-based distribution (Equation (4)) and TC is the rate per m$^3$ traded, with $0 \leq TC \leq MP_t / WM_t^i - WR_t^i$.

$$MP_t = \sum_i \pi_t^i \left( X_t^{i*} \left( WM_t^i \right) \right) - \sum_i \pi_t^i \left( X_t^{i*} \left( WR_t^i \right) \right) - (WM_t^i - WR_t^i) \times TC \qquad (8)$$

The fixed cost component in the transaction was defined as the average amount of water exchanged per transaction (AWT) at the seasonal level, which was obtained from the WUR transactions during the years 2015–2017. The market was simulated by adding a cost (penalty) to each transaction, as shown in Equation (9), where $WT_t$ indicates the water transactions and is calculated as total water traded divided by AWT.

$$MP_t = \sum_i \pi_t^i \left( X_t^{i*} \left( WM_t^i \right) \right) - \sum_i \pi_t^i \left( X_t^{i*} \left( WR_t^i \right) \right) - WT_t \times TC \qquad (9)$$

*2.3. Model Validation*

In order to validate the model results, the predictions were compared to real estate values of agricultural land in the area. Then, the predicted values were compared with the weighted values of land with water in the area offered for sale on real estate web sites. Perpetual values of land and water rights were calculated based on the discounted cash flow assessment methodology according to the marginal product method. The discounted cash flow was calculated as the quotient of annual profits over the discount rate, as shown in Equation (10):

$$V = \sum_{t=1}^{\infty} \frac{C}{(1+r)^t} = C/r \qquad (10)$$

where $C$ is the annual profit gross in US dollars ($), corresponding to the weighted profit per hectare in the study area for an average of 30 irrigation seasons. The weights of each farm size were based on the property stratification in the zone, considering that farms smaller than 12 hectares cover 12.1% of the area, farms between 12 and 100 hectares account for 55.6%, and farms larger than 100 hectares account for 32.3% of the total. $r$ is the discount rate that ranges from the proposed 3% for land [43] to the proposed 7% for water [44,45].

*2.4. Ñuble River WUR Market Study*

Information on water rights transactions between 2015–2017 was obtained from the Water Property Registry (RPA) archives at the Real Estate Registry Office (CBR) of the cities of San Carlos and Chillán. The Water Code requires that water rights be registered as property or real estate. Therefore, RPAs were a reliable source of information for assessing the functioning and dynamism of the water rights market in Chile. This information included data on the amount of water traded, the location on the canal and the water transaction price. The weighted average price, number of transactions, and the quantity of water traded were calculated. Market depth (MD) and market activity (MA) were estimated from Equations (11) and (12), where TS is the amount of annual water shares traded, AS is the number of all shares, CS is the number of sellers or transactions and PS is the amount of all possible sellers or WUR owners. The analyses considered the entire river system, separated by banks (northern and southern banks). Only one out of the 667 studied transactions was between different canals.

$$MD = TS/AS \times 100 \tag{11}$$

$$MA = CS/PS \times 100 \tag{12}$$

## 3. Results

When analyzing the results presented below, it is important to address them with a broad perspective. Although in this case they respond to a market system resulting from the institutional and legal arrangement of Chile, the phenomena and processes associated with water distribution and allocation such as transaction and information costs are present under any water management system.

### 3.1. Estimation of Benefits

Table 1 summarizes the benefits predicted by the model for each type of crop and includes the cultivated surface area, water demand (crop and total irrigation system), and expected benefit (per hectare and per water volume). Fruit orchards have the highest profitability of water as they use close to 5% of the total water demand and account for up to 40% of the total profits in the area (sum of the seasonal profits of the 60,000 hectares of the study area). Other profitable crops include seeds and vegetables, which account for less than 5% of total water consumption and provide up to 20% of the profits of the agricultural industry in the region. Conversely, grassland and forage have the worst water use profitability, with a potential water consumption close to 40% of the total demand and less than 10% of total profits.

When estimating the overall benefits of water use in the irrigated agricultural fields, the weighted average profit was used considering each crop type available in the area. The scope of analysis did not include adding newly irrigated fields or increasing water availability. Instead, the analysis focused on increasing the economic benefit of the agricultural area by improving water management and allocation within the irrigation network. This information (Table 1) was used as a baseline for estimating the benefits of a water market along with water supply and demand curves for each hydrological year.

The benefits of managing water as an economic good depend largely on its value; the higher the value of the good, the greater the benefits of its trade or mobility. Figure 4 shows the total annual profits of the ÑRIS under current and perfect water market conditions, as a function of water availability (occurrence probability). Dry years (drought) are represented on the left side of the chart, while wet years are represented on the right side. For years with lower water availability, benefits are greater when water is allocated as an economic good due to the significant profit differences between the ideal (strategic) and current distribution approaches, reaching up to $20 million for the driest year. However, for years with higher water availability (close to the total system demand), the benefits are lower. The average net annual return of the system under current conditions reaches $30.3 million, with a standard deviation of $5.5 million, while the average net annual return under perfect market conditions reaches $37.8 million, with a standard deviation falling to $0.7 million. Therefore, the mean annual system cost due to market inefficiencies is $7.5 million, which represents 25% of total ÑRIS profits.

Improving the market conditions for the ÑRIS not only increases the total profit of agricultural production (4.2% to 5.3%) but also decreases the risk of the activity, from 18.2% to 2.0% (see Figure 4). These improvements increase land values [46] through improvements in water transport and distribution infrastructure and water management and allocation systems, which are crucial for decreasing transaction costs (see Figure 5).

A flexible distribution system allows for policies that consider not only economic factors but also effects on the water footprint, as found in a parallel study [47].

**Table 1.** ÑIRS hydro-economic model. Surface area, water demand, and profitability per crop type. Seasonal Data.

| Crops and Vegetables | Surface Area (ha) | Water Demand (m³/ha) | Total System Demand (m³) | Benefits $/ha | Benefits $/Ml [a] |
|---|---|---|---|---|---|
| Cereals and crops | | | | | |
| Rice | 1571.4 | 13,921 | 21,875,423 | 845 | 61 |
| Wheat | 9072.0 | 12,534 | 113,717,021 | 372 | 30 |
| Oats | 1581.6 | 12,858 | 20,337,300 | 75 | 6 |
| Grain corn | 4632.0 | 12,425 | 57,554,141 | 686 | 55 |
| Potato | 134.9 | 11,658 | 1,572,335 | 1677 | 144 |
| Bean | 1480.0 | 10,995 | 16,273,539 | 908 | 83 |
| Other crops | 521.0 | 7959 | 4,146,836 | 1090 | 137 |
| Industrial crops | | | | | |
| Chicory | 283.7 | 7244 | 2,054,908 | 1110 | 153 |
| Beet | 2445.6 | 12,743 | 31,166,521 | 1070 | 84 |
| Others | 310.9 | 13,202 | 4,105,334 | 1090 | 83 |
| Seed and vegetables | | | | | |
| Corn seed | 754.2 | 10,045 | 7,576,431 | 2117 | 211 |
| Sunflower seed | 575.0 | 8540 | 4,910,781 | 3071 | 360 |
| Green pea | 57.7 | 6716 | 387,426 | 737 | 110 |
| Corn | 552.6 | 10,647 | 5,883,010 | 706 | 66 |
| Asparagus | 470.5 | 8173 | 3,845,819 | 2013 | 246 |
| Fava bean | 43.0 | 3257 | 140,129 | 938 | 288 |
| Green bean | 141.5 | 7148 | 1,011,364 | 1003 | 140 |
| Tomato | 110.1 | 14,394 | 1,584,947 | 3967 | 276 |
| Carrot | 107.7 | 8950 | 963,759 | 1743 | 195 |
| Zucchini | 207.0 | 8304 | 1,719,054 | 5290 | 637 |
| Fruits | | | | | |
| Blueberry | 1101.8 | 6699 | 7,380,857 | 3681 | 549 |
| Raspberry | 282.7 | 10,397 | 2,939,783 | 4040 | 389 |
| Strawberry | 185.2 | 5869 | 1,087,138 | 6551 | 1116 |
| Cherry | 425.0 | 9449 | 4,015,870 | 8010 | 848 |
| Kiwi | 417.8 | 9713 | 4,058,103 | 1980 | 204 |
| Apple | 520.6 | 11,314 | 5,889,821 | 3009 | 266 |
| Walnut | 127.8 | 14,721 | 1,881,874 | 3580 | 243 |
| Indoor fruit | 21.2 | 2489 | 52,842 | 441 | 177 |
| Others | 152.6 | 7866 | 1,200,435 | 4407 | 560 |
| Table grape | 340.9 | 8944 | 3,048,733 | 8244 | 922 |
| Meadows and forage | | | | | |
| Corn silage | 1081.8 | 11,577 | 12,524,202 | 124 | 11 |
| Alfalfa | 781.8 | 20,947 | 16,377,196 | 124 | 6 |
| Clover | 3253.0 | 21,026 | 68,398,327 | 124 | 6 |
| Mixed meadow | 3013.4 | 21,091 | 63,557,084 | 124 | 6 |
| Natural meadow | 12,262.4 | 21,088 | 258,597,391 | 124 | 6 |
| Other meadows | 1501.2 | 21,000 | 31,526,624 | 124 | 6 |
| Weighted Total | 50,521.5 | 14,367 | 783,362,354 | 770 | 50 |

[a] Megaliter (1000 m³). Data sources: 2007 National Agricultural Census for crop area and crop categorization, [37] for water consumption, and Office of Agrarian Studies and Policies (ODEPA) production sheets for economic benefits. The remaining figures are calculations between columns.

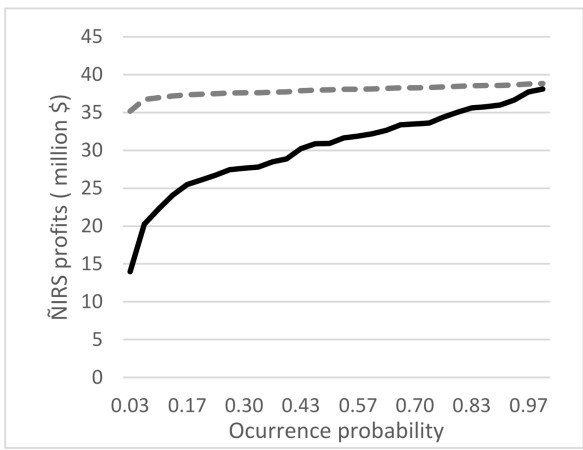

**Figure 4.** Economic profits of the ÑRIS. Hydrologic occurrence probability for market profits under current (solid line) and perfect (dotted line) market conditions.

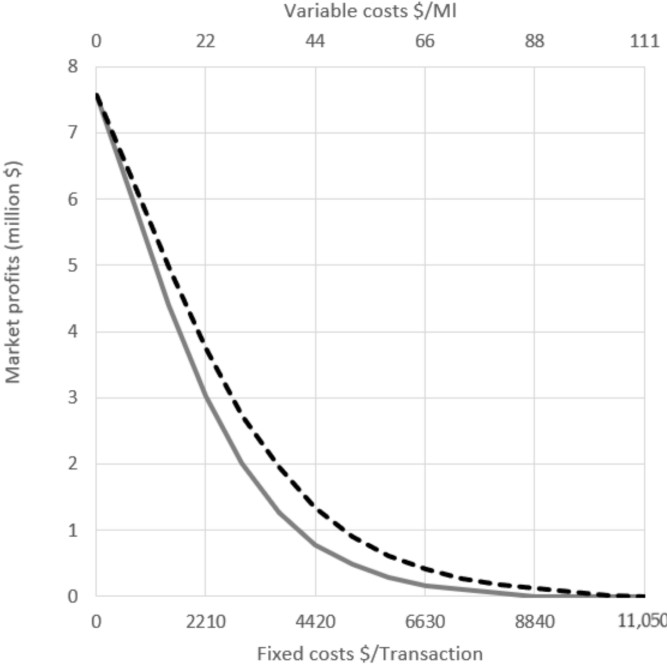

**Figure 5.** Transaction costs and market profits. Relationship between fixed (solid line) and variable (dotted line) transaction costs and market benefits.

*3.2. Transaction Costs*

It is important to study the effects of transaction costs, since in the near future, each farmer may be able to store water and request it from the reservoir. Figure 5 shows the effects of transaction costs on potential water market benefits (current system costs) for a range of transaction costs, both variable and fixed. With transaction costs of $22.1/Ml (variable) and $1800/transaction (fixed), the benefits of a water market are reduced by 50%.

In addition, an increase of 1 dollar in variable transaction costs per megaliter results in a decrease of $70,727 in market benefits. Similarly, an increase of 1 dollar in fixed transaction costs reduces market benefits to $867. The magnitude of the effects of transaction costs on the net benefits of the market [34] highlights the importance of an institutional framework to reduce these costs, as a reduction will not occur by itself.

### 3.3. Validation

Once the system costs of market inefficiencies were known, the model was validated by predicting the commercial value of the agricultural land in the area. Since the marginal product model uses land and water profits to value water, the same concept was used to predict the current market value of land in the Ñuble River irrigated area. Figure 6 shows the market values and predictions for the cost of irrigated land adjusted by the discounted cash flow assessment method. It is observed that the value decreases as the required rate of return on capital increases, with an intersection at about 4.2%.

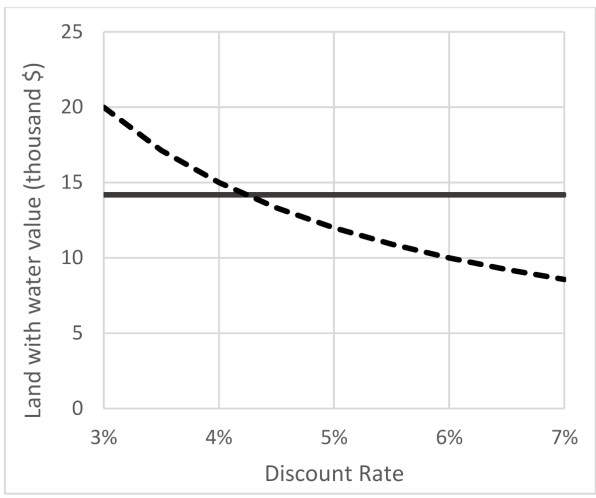

**Figure 6.** Validation and discount rate. Relationship between the market (solid line) and model (dotted line) predicted value of land with water.

The discount rate affects the model capabilities of assessing the profitability of off-farm irrigation investments in the area. When establishing a discount rate for the irrigation system, a wide range of values may be considered. While a blueberry farmer's internal rate of return (IRR) on water and land investments reaches 20.2%, that of a wheat farmer barely exceeds 2.0%. In addition, when comparing the required rate of return on capital of the ÑRIS (4.2% for the weighted average of the crops) with the Capital Assessment Price Model (CAPM) [48] (5.4% for a five-year timeframe), there was a difference of 1.2 percentage points, while the social discount rate used in Chile is 6% (law 18450), which is used to promote private investment projects in irrigation.

### 3.4. Ñuble River WUR Market Study

The San Carlos CBR maintains the water records for the northern bank of Ñuble River, where 16,665.77 shares are distributed among 37 canals, which accounts for 78.5% of all the river shares. The survey of this CBR resulted in a database with 580 records, with a total of 1533.78 shares traded. Meanwhile, the Chillán CBR maintains the water records for the southern bank of the Ñuble River, where 4484.93 shares are distributed among 14 canals, representing 21.1% of the river shares. A total of 87 records were obtained from this CBR, with a total of 356.37 shares. Table 2 shows that the northern bank of the river has a greater mean share value (price) compared to the southern bank. This also occurs for price dispersion and market depth. By contrast, greater market activity occurred in the southern bank compared to the northern riverbank.

**Table 2.** Ñuble River WUR market study. Number of transactions, quantity traded (shares), market depth and activity, mean price, weighted price and standard deviation of Ñuble River WUR separated by section (2015–2017).

| Section | Number of Transactions | Quantity Traded (Shares) | Market Depth | Market Activity | MEAN PRICE ($) | Weighted Price ($) | Unit Price STD ($) |
|---|---|---|---|---|---|---|---|
| Northern bank | 580 | 1533.78 | 3.07 | 4.15 | 5272 | 3628 | 7480 |
| Southern bank | 87 | 356.37 | 2.65 | 4.66 | 3229 | 1929 | 3190 |
| Entire river | 667 | 1891.31 | 2.98 | 4.21 | 5005 | 3308 | 7101 |
| Canal average | 13.34 | 37.83 | 3.57 | 7.69 | 4130 | 3227 | 4134 |

## 4. Discussion

Share price dispersion was greater than the average share price, demonstrating market inefficiencies. However, the price dispersion decreased within each channel, supporting the idea that each channel is a separate market and is consistent with the analysis of transaction costs associated with changes in proportional flow dividers and their harmful effect on the market. A comparison of market depth values with those found by [49] for the Maipo and Mapocho Rivers showed similar results, a situation that reveals similar levels of WUR market development in the different basins. Meanwhile, the high price dispersion in the WUR market found in the La Paloma system of the Limarí River [50] demonstrates that price dispersion results not only from a distribution infrastructure limitation, but also from the legal and institutional framework [21].

The Real Estate Registry Office does not provide updated information on water transactions: (i) the only prices that can be easily accessed are from previous years, and current information is not readily available; (ii) the confusing and ambiguous format in which transactions are recorded does not have a fixed structure [30]; and (iii) because the transactions are eligible for taxation based on the sale price and not the quantity traded, there is a tendency to undervalue the transactions to avoid paying higher taxes [20]. It would be beneficial to create a new institution that gathers information on water transactions, suppliers, and demanders in order to make the WUR market more efficient and transparent.

Ref. [51] stated that the agricultural benefits of improved water management would outweigh the costs of infrastructure changes. Although the study only considered the increased agricultural (operational) profitability of the land, it should be noted that improvements in water transportation and distribution infrastructure and in water management and allocation increase the real estate value of the land [46]. The degree of organization and adaptability of the irrigation system to climate variability is an attribute that could improve the real estate value of land in the area.

The average share prices obtained for the transactions during 2015–2017 for each of the canals of the Ñuble River are shown in Figure 7. There is a great difference in WUR market behavior among the canals, with greater water rights prices on the northern bank compared to the southern riverbank. Only one intercanal transaction was observed during the three-year study period. Therefore, there is a market for each canal rather than for the entire irrigation system, each market has a limited number of participants, which is not consistent with the assumption of a perfect market with a larger number of participants.

The market for water rights by channel was studied in order to define strategies for prioritizing the construction of replacements for the water distribution structures carried out by the Supervisory Board. To the knowledge of the authors, this type of study is one of the pioneer studies in Chile. In addition, the results of this research motivated the decision for an extensive development strategy per channel, which is currently renewing all the relevant distribution structures in a single channel, and then moving to the next channel and so on instead of replacing the entire distribution infrastructure at once (main, secondary and tertiary distribution infrastructure). This strategy is more suitable for the current management dynamics of the board.

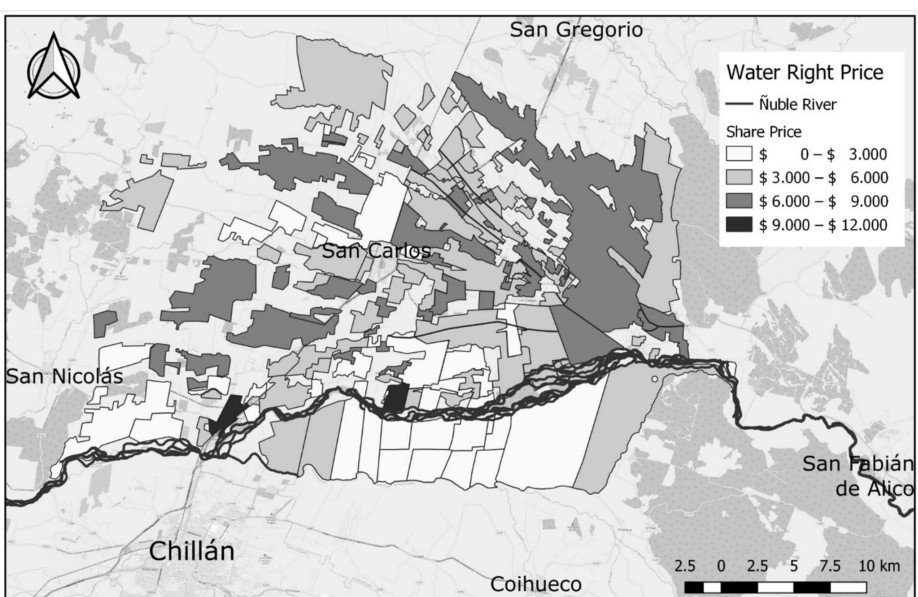

**Figure 7.** Geographic price dispersion of water use rights (WUR) in ÑRIS. Share price for water rights by canal in the Ñuble River irrigation system.

Actual economic yields observed in current water markets in Chile are far from their potential yields [10]. Given the lower availability of flowing water due to climate variability [11,35], there is a need for more competitive water markets and a more efficient use of this limited resource. The findings regarding the limitations of the Ñuble River water market can be extended to other water markets in Chile and other countries with similar challenges in terms of rigid infrastructure and climate variability. In addition, the proposed model allows for an economic assessment of off-farm irrigation investments that leads to better management of water, a scarce resource.

## 5. Conclusions

The WUR market in Chile has a fundamental problem. It promotes efficiency in trading water rights rather than promoting efficient use of water, considering that it is a limited resource under climate variability and drought. Therefore, the WUR market is an ineffective strategy for water allocation on a regular basis. In addition, the water market alone does not maximize the long-term benefit of water use for overall agricultural production in the Ñuble region, since it prioritizes among existing higher-value uses, rather than increasing the value of use on individual properties. Therefore, it must be complemented with strategic water planning for territorial adaptation to the most appropriate and profitable crops for conditions of climate variability. However, the lack of an efficiently operating water market in the area generates an average annual cost of the system that was estimated at $7.5 million, which represents 25% of total ÑRIS profits. Therefore, strategic planning should be aimed at promoting flexibility in water distribution at the canal level and more profitable crops suitable for the climatic variability in the region.

Further work may include the effects of surface and groundwater interaction to integrate them into the economic model for improving water management scenarios. In addition, georeferencing the crops of each farm, canals, and their distribution works would allow a shift to a spatial model of water resources management that includes the supply, demand, distribution, and storage of water resources in the area.

**Author Contributions:** Conceptualization, investigation and formal analysis by B.B., J.J., and J.L.A.; methodology and formal analysis by B.B. and J.L.A. Writing—original draft preparation and editing, B.B., J.J., and J.L.A. All authors have read and agreed to the published version of the manuscript.

**Funding:** This research was supported by the project ANID/FONDAP/15130015.

**Institutional Review Board Statement:** Not applicable.

**Informed Consent Statement:** Not applicable.

**Data Availability Statement:** No new data were created or analyzed in this study. Data sharing is not applicable to this article.

**Acknowledgments:** Junta de Vigilancia del Río Ñuble, Centro de Recursos Hídricos para la Agricultura y la Minería CRHIAM ANID/FONDAP/15130015, Jesus Christ, water.

**Conflicts of Interest:** The authors declare no conflict of interest. The funders had no role in the design of the study; in the collection, analyses, or interpretation of data; in the writing of the manuscript, or in the decision to publish the results.

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
