# Peer review of "How Much Does Water Management Cost? The Case of the Water Market in the Ñuble River of South-Central Chile"

_water, doi:10.3390/w13030258_

Round 1

Reviewer 1 Report

Review of “How much does water management cost? The case of the water market in the Nuble River of south-central Chile”

Overall comment

This paper attempts to show that the current water management system along the Nuble River is not efficient. To do so, the authors looked at transaction cost and benefit estimation by crops. Much information presented here sounds original and promising. However, at this current condition, I think it is premature to publish it. I recommend that the authors make some major revisions. For example, I do not find that the authors correctly understand water rights and water market issues in terms of jurisprudence and administrative frameworks. Please also have someone to check grammar. There are some missing “the” and other gremlins that an editorial service can easily pick up for them. Below I made some specific comments to aid their revision efforts in the future.

Introduction

Page 1, lines 37-38: The authors claim that “demand management has led to a legal separation of water management from land management” in the UK, Australia, the U.S. and Canada. This is a dubious and erroneous claim. The riparian right regime that recognizes water rights of riparian owners still exist in various forms in the U.S., Canada, Australia, and the U.K. The U.S., Canada, and Australia use both riparian rights and prior appropriation. The authors need to cite more proper sources of information to make this sweeping claim. There are many works on water rights development in these countries.

Page 1, lines 40-41: The authors claim that Chile’s administrations created “one of the most liberal regulatory frameworks.” I am not clear what they mean here. This sentence sounds like a topic sentence for this paragraph, but nothing about this “liberal regulatory frameworks” is discussed. Later, the authors explain about challenges of water governance in the Nuble watershed, including insufficient streamflow and water market information, poor record keeping, and “rigid distribution infrastructure.”

Page 2, lines 48-50: The authors now seem to switch a topic to a scenario analysis. If this analysis is important for this paper, it is very helpful here to explain why this perfect scenario analysis is important and what the past studies have done on this topic.

1.3 Study area

Page 4, lines 135-138: The authors write: “The Nuble River Water Board currently manages the water supply of the canal network, among which the streamflow of the river is distributed via proration proportional to each canal and user’s ownership of water shares.” This point can be enhanced by incorporating what the authors claim in their “Discussion.” See my comments on your Discussion below.

1.4 Objectives

Page 4, lines 160-161: The authors claim that their objective was to “to contribute to the literature by providing a methodology for estimating the opportunity cost of maintaining a fixed traditional irrigation canal system under climate change.” So far, I am not convinced that the authors did so. Explain more about the necessity of this “methodology” and explain if this methodology is original and contribute to a scholarship. Also, information about climate change is not clear yet. The sections for results and conclusion need to elaborate more on climate change.

3.1 Estimation of benefits

Page 7, lines 277-284: Before discussing this paragraph, it is helpful to explain, perhaps in the methodology, what types or categories of crop or cropland the authors used (Table 1 uses somewhat different crop categories). Here the authors seem to have categories of fruit orchards, seedbeds, vegetables, grassland and forage. Did they follow a certain national census standard to categorize crop/cropland this way? Here it is also helpful for readers to know what the authors mean by “the total profit in the area” (line 280) and when (e.g., year or month) this was so. Earlier the authors pointed out about the shortage of irrigation water in summer and fall in this river, but here, I am not yet clear if the authors are actually addressing the seasonal fluctuation of water flows.

Table 1: This table does not indicate the sources of the data. Did the authors take information from any statistical services that were available or is this table from their original enumeration and analysis efforts? Either way, the authors need to clarify the source. Also, the information about tomato is missing. Or should the area for “fresh consumption” be for tomato? Here tomato should use upper case “T” to be persistent with other categories above and below.

  1. Discussion

Page 11, lines 368-369: The authors point out about “a great difference in WUR market behavior among the canals.” Please mention this point in 1.3 “Study area.” Also, the authors now reveal that “there is a market for each canal rather than for the entire irrigation system.” This information sounds important to be mentioned either in “1.3 Study area” or methodology, in which they explain about their water market analysis. In fact, much of the information discussed in this section should be explained in methodology.

  1. Conclusion

This conclusion sounds superficial. The authors clearly explain their major findings here.

Author Response

Responses to Reviewer: 1

Page 2, lines 50-54: The sentence needs to be amended to make it more readable. It is too long.

Answer:

Original sentence:

This study can be of global interest since it is a place where the water management was separated from the land by means of a water market and is addressing a water allocation problem [14] by a supply management approach, which includes a water reservoir that is under technical assessment

Modification [Page 2, lines 61-63]

This study may  be of global interest since it addresses a water allocation problem [14] using a supply management approach, which includes a water reservoir that is under technical assessment.

Page 4, lines 144-149: With regard to the topic addressed in the article, the description of the canals is too detailed.

Answer:

Original text:

At their nodes (Figure 2-a), the canals have a distribution box called proportional flow dividers, which are critical flow hydraulic structures that divide the variable streamflow of a canal into a fixed proportion, according to water rights, by using rigid structures, which are common in Chile´s irrigation canals. They are essentially composed of (Figure 2-b) a chamber (D) that provides critical flow to the inlet streamflow (A) and a sheet (E) that divides the streamflow into instream (B) and outlet (C) in proportion to its transverse location.

Modification [Page 4, lines 166-168]

The description was summarized as follows: “At their nodes (Figure 2), the canals have proportional flow dividers, which are critical flow hydraulic structures that divide the incoming variable streamflow of the canals into different outlets at a fixed proportion, according to water rights (Figure 2-b).”

There is no need for a separate chapter to present objectives. They should be formulated at the end of the 'Introduction' chapter.

Answer: The objectives paragraph was added to the introduction; see in Page 2, lines 66-72.

What does 'P' in equation 1 mean? This parameter needs to be clarified.

Answer: Pi represents the price of the crop; the explanation was added to the text. See Page 6, line 198.

page 5, line 195: The authors refer to the parameter 'efficiency ???',

which was not defined before. This parameter should be explained earlier in the text.

Answer: This nomenclature was removed and now it is listed as application efficiency of the irrigation technique on page 6 line 208.

In discussing the results, the water demand of crops could relate to the water footprint. Have the authors tried to address this issue from this perspective?

Answer: Very interesting concept, which was investigated in a parallel study by Novoa et al. (2019) and  cited in the text of the manuscript on page 9, lines 327-329

Novoa, V., Ahumada-Rudolph, R., Rojas, O., Sáez, K., de la Barrera, F., & Arumí, J. L. (2019). Understanding agricultural water footprint variability to improve water management in Chile. Science of the Total Environment, 670, 188–199. https://doi.org/10.1016/j.scitotenv.2019.03.127

In 'conclusions' there is no reference to test results. It is more a summary than a conclusions.

Answer: The conclusion was improved.

Reviewer 2 Report

Specific comments

Comments related to the content of the article are the following:

  1. Page 2, lines 50-54: The sentence needs to be amended to make it more readable. It is too long.
  2. Page 4, lines 144-149: With regard to the topic addressed in the article, the description of the canals is too detailed.
  3. There is no need for a separate chapter to present objectives. They should be formulated at the end of the 'Introduction' chapter.
  4. What does 'P' in equation 1 mean? This parameter needs to be clarified.
  5. page 5, line 195: The authors refer to the parameter 'efficiency ???', which was not defined before. This parameter should be explained earlier in the text.
  6. In discussing the results, the water demand of crops could relate to the water footprint. Have the authors tried to address this issue from this perspective?
  7. In 'conclusions' there is no reference to test results. It is more a summary than a conclusions.

Author Response

Responses to Reviewer: 2

Overall comment

This paper attempts to show that the current water management system along the Nuble River is not efficient. To do so, the authors looked at transaction cost and benefit estimation by crops. Much information presented here sounds original and promising. However, at this current condition, I think it is premature to publish it. I recommend that the authors make some major revisions. For example, I do not find that the authors correctly understand water rights and water market issues in terms of jurisprudence and administrative frameworks. Please also have someone to check grammar. There are some missing “the” and other gremlins that an editorial service can easily pick up for them. Below I made some specific comments to aid their revision efforts in the future.

Answer: The introduction section was modified to better explain the institutional framework of water rights in Chile. The grammar was checked throughout the manuscript.

Introduction

Page 1, lines 37-38: The authors claim that “demand management has led to a legal separation of water management from land management” in the UK, Australia, the U.S. and Canada. This is a dubious and erroneous claim. The riparian right regime that recognizes water rights of riparian owners still exist in various forms in the U.S., Canada, Australia, and the U.K. The U.S., Canada, and Australia use both riparian rights and prior appropriation. The authors need to cite more proper sources of information to make this sweeping claim. There are many works on water rights development in these countries.

Answer: The statement “demand management has led to a legal separation of water management from land management” was removed from the text. An attempt was made to use the concept of separation of water and land as a synonym for demand management. What was important was the focus on demand management and water mobility independent of land as in the water bank in California or the water markets in Australia, rather than the legal nature of the rights. Modification Page 1, line 37.

Page 1, lines 40-41: The authors claim that Chile’s administrations created “one of the most liberal regulatory frameworks.” I am not clear what they mean here. This sentence sounds like a topic sentence for this paragraph, but nothing about this “liberal regulatory frameworks” is discussed. Later, the authors explain about challenges of water governance in the Nuble watershed, including insufficient streamflow and water market information, poor record keeping, and “rigid distribution infrastructure.”

Answer: The statement “one of the most liberal regulatory frameworks” was removed from the text and the paragraph was modified see Page 2, lines 39-45.

Page 2, lines 48-50: The authors now seem to switch a topic to a scenario analysis. If this analysis is important for this paper, it is very helpful here to explain why this perfect scenario analysis is important and what the past studies have done on this topic.

Answer: An explanation of the use of a perfect market was included in the text, as can be seen on Page 2, lines 55-61. With respect to previous studies, to the knowledge of the authors there are no previous studies that address the topic in the area.

1.3 Study area

Page 4, lines 135-138: The authors write: “The Nuble River Water Board currently manages the water supply of the canal network, among which the streamflow of the river is distributed via proration proportional to each canal and user’s ownership of water shares.” This point can be enhanced by incorporating what the authors claim in their “Discussion.”

See my comments on your Discussion below.

Answer: To the authors' knowledge, never in Chile had the water market been studied on the basis of individual canals rather than as a complete river system or a section of a river. As part of the development of the project to modernize the infrastructure of the Ñuble River canals, it was

necessary to decide how to prioritize resources. The options consisted of prioritizing the change of works in the gates with the highest flow rate in each of the canals or improving the water distribution system at canal level sequentially. This research was innovative in that it provided an assessment of the potential benefits of this distribution approach at canal level for the Ñuble River irrigation network. In addition. as a result of this research, the Nuble River Water Board adopted a strategy of seeking the implementing automatic gate systems for each canal.

1.4 Objectives

Page 4, lines 160-161: The authors claim that their objective was to “to contribute to the literature by providing a methodology for estimating the opportunity cost of maintaining a fixed traditional irrigation canal system under climate change.” So far, I am not convinced that the authors did so. Explain more about the necessity of this “methodology”

and explain if this methodology is original and contribute to a scholarship. Also, information about climate change is not clear yet.

The sections for results and conclusion need to elaborate more on climate change.

Answer: This research aims to contribute to the literature by estimating the opportunity cost of maintaining a fixed traditional irrigation canal system under climate variability. This was corrected in the text. See Page 2, line 67 and Page 11, line 384.

3.1 Estimation of benefits

Page 7, lines 277-284: Before discussing this paragraph, it is helpful to explain, perhaps in the methodology, what types or categories of crop or cropland the authors used (Table 1 uses somewhat different crop categories). Here the authors seem to have categories of fruit orchards, seedbeds, vegetables, grassland and forage. Did they follow a certain national census standard to categorize crop/cropland this way? Here it is also helpful for readers to know what the authors mean by “the total profit in the area” (line 280) and when (e.g., year or month) this was so. Earlier the authors pointed out about the shortage of irrigation water in summer and fall in this river, but here, I am not yet clear if the authors are actually addressing the seasonal fluctuation of water flows.

Answer: The total profits in the area are listed by season, as explained on Page 8, line 296: “sum of the seasonal profits of the 60,000 hectares of the study area.” The market simulation deals with the interseasonal variation of water flows and for the scope of the study the seasonal-step model was developed. The reasons for this seasonal variation are explained on Page 4, lines 149-153 and addressed in the simulation (Page 5, lines 196-197). The categories for crops were indicated in the footnote of Table 1 in Page 9, line 307-308

Table 1: This table does not indicate the sources of the data. Did the authors take information from any statistical services that were available or is this table from their original enumeration and analysis efforts? Either way, the authors need to clarify the source. Also, the information about tomato is missing. Or should the area for “fresh consumption” be for tomato? Here tomato should use upper case “T” to be persistent with other categories above and below.

Answer: The data sources were indicated on Page 5, lines 197-199 (VII National Agriculture and Livestock Census of Chile and 2013 Office of Agrarian Studies and Policies (ODEPA) Production Cost reports) and in Page 6, lines 206-208 (MdeA Consultores Ltda. (2009). Evaluación Agro-Social del Embalse La Punilla).

In addition, the source of information was added to the footnote of Table 1; see Page 9, lines 307-309. “Tomato” in Table 1 was corrected to be consistent with the other categories.

Discussion

Page 11, lines 368-369: The authors point out about “a great difference in WUR market behavior among the canals.” Please mention this point in

1.3 “Study area.” Also, the authors now reveal that “there is a market for each canal rather than for the entire irrigation system.” This information sounds important to be mentioned either in “1.3 Study area”

or methodology, in which they explain about their water market analysis.

In fact, much of the information discussed in this section should be explained in methodology.

Answer: This correction was included in the text.

Conclusion

This conclusion sounds superficial. The authors clearly explain their major findings here.

Answer: The conclusion was improved.
